# REASSURED diagnostics at point-of-care in sub-Saharan Africa: A scoping review

**Boitumelo Moetlhoa**[1]*, **Kuhlula Maluleke**[2], **Evans M. Mathebula**[2,3], **Kabelo Kgarosi**[4], **Siphesihle R. Nxele**[1], **Bonolo Lenonyane**[1], **Tivani Mashamba-Thompson**[1]

**1** Faculty of Health Sciences, University of Pretoria, Pretoria, South Africa, **2** Faculty of Health Sciences, School of Health Systems and Public Health, University of Pretoria, Pretoria, South Africa, **3** Medical and Scientific Affairs, Rapid Diagnostics, Infectious Diseases Emerging Markets, Abbot Rapid Diagnostics (Pty) Ltd, Sandton, South Africa, **4** Faculty of Health Sciences, Department of Library Services, University of Pretoria, Pretoria, South Africa

* u22029992@tuks.co.za

## Abstract

Point-of-care (POC) diagnostics that meet the REASSURED criteria are essential in combating the rapid increase and severity of global health emergencies caused by infectious diseases. However, little is known about whether the REASSURED criteria are implemented in regions known to have a high burden of infectious diseases such as sub-Saharan Africa (SSA). This scoping review maps evidence of the use of REASSURED POC diagnostic tests in SSA. The scoping review was guided by the advanced methodological framework of Arksey and O'Malley, and Levac et al. We searched the following electronic databases for relevant literature: Scopus, Dimensions, ProQuest Central, Google Scholar, and EBSCO-host (MEDLINE, CINAHL, as well as AFRICA-WIDE). Two reviewers independently screened abstracts and full-text articles using the inclusion criteria as reference. We appraised the quality of the included studies using the mixed-method appraisal tool (MMAT) version 2018. We retrieved 138 publications, comprising 134 articles and four grey literature articles. Of these, only five articles were included following abstract and full-text screening. The five included studies were all conducted in SSA. The following themes emerged from the eligible articles: quality assurance on accuracy of REASSURED POC diagnostic tests, sustainability of REASSURED POC diagnostic tests, and local infrastructure capability for delivering REASSURED POC diagnostic tests to end users. All five articles had MMAT scores between 90% and 100%. In conclusion, our scoping review revealed limited published research on REASSURED diagnostics at POC in SSA. We recommend primary studies aimed at investigating the implementation of REASSURED POC diagnostic tests in SSA.

## Introduction

Major scientific developments to strengthen health systems, improve sustainability, and promote human health have been made over the years. Yet many countries, including those in sub-Saharan Africa (SSA), are still confronted by long standing emerging and re-emerging

**Data Availability Statement:** Data provided as part of submission.

**Funding:** The authors received no specific funding for this work.

**Competing interests:** The authors declare no competing interests exist.

infectious disease threats [1,2]. This may be due to the region having low capacity for risk management of disease epidemics which is mainly caused by lack of adequate resources for early detection, identification, as well as timeous response [1]. Early diagnosis of infectious diseases such as Human Immunodeficiency Virus (HIV), chlamydia, and tuberculosis (TB) reduces transmission caused by untreated cases [3]. Thus, high-quality diagnostic tests are needed for surveillance and to control the rapid spread of infectious diseases, as well as to monitor communicable disease risk factors such as blood glucose and cholesterol levels [3,4]. Point-of-care (POC) diagnostic tests are medical tests conducted outside a clinical laboratory, near the site of the patient or patient bed site and provide affordable, reliable as well as rapid diagnosis [5,6]. POC diagnostic tests do not require specialized training and are often performed by non-laboratory personnel such as nurses, physicians, and patients [5]. POC diagnostic tests are commonly used where the turnaround time for patient results is short or in areas where central laboratory testing is unavailable [7]. POC diagnostics tests thus offer better accessibility to diagnostic testing in remote areas or patient homes [8].

The lack of high-quality diagnostics in resource limited healthcare systems causes significant delays in surveillance, detection, control, and prompt treatment of infectious diseases. To address this challenge, the World Health Organization Special Program for Research and Training in Tropical Diseases (WHO/TDR) introduced the ASSURED (Affordable, Sensitive, Specific, User-friendly, Rapid, Equipment-free, Delivered) criteria for ideal tests to improve access to diagnostic tests for all levels of healthcare systems [9].

The rapid increase and severity of global health emergencies caused by infectious diseases have exposed the need for next-generation devices for POC tests, which could catalyze transformation in healthcare delivery [10,11] This phenomenon has also led to a review of the ASSURED criteria [11]. Considering advances in digital technology, the ASSURED criteria were upgraded to include real-time connectivity and ease of sample collection, subsequently becoming the REASSURED criteria as an improved benchmark for ideal diagnostic tests [9,12] The WHO further published a model essential diagnostics list to keep diagnostics on the agenda for global health. Despite the availability of this list, a survey revealed limited availability of these diagnostic tests at primary-care level in low-and middle-income countries [13]. Lack of access to POC diagnostic tests has negatively affected the diagnostics field in the influx of care for many common infectious diseases [13]. Diagnostics that meet the REASSURED criteria have been shown to incorporate new technological tools [9], which provide real-time quality control testing and treatment. Real-time diagnostics tests allow for the timely electronic distribution of information to widely dispersed healthcare professionals, who can use this information to collaborate and provide the best medical advice and treatment to patients. The early adoption of next generation POC diagnostics by SSA countries may benefit and enhance progress towards Universal Health Coverage as well as the United Nations Sustainable Development Goal 3 (SDG) 3 [10].

REASSURED POC diagnostic tests simplify the collection and processing of samples, which would otherwise be processed by a professional pathologist [9,12]. This scoping review maps the evidence of the implementation of REASSURED POC diagnostic tests in SSA. We anticipate that our results will help guide future research to improve the implementation of REASSURED POC diagnostic tests in SSA.

## Methods

### Study design

The protocol for this scoping review was developed a priori and registered with the Open Science Framework (OSF) under the title: "REASSURED diagnostics for public health outcomes

in sub-Saharan Africa: A scoping review protocol." The protocol is accessible via this link: https://archive.org/details/osf-registrations-zs3pb-v1.

This scoping review was guided by the methodological framework of Arksey and O'Malley [14], which was further advanced by Levac et al. [14] to include quality assessment of findings. Based on this framework, we conducted the scoping review according to the following five stages: identifying the research question, identifying relevant studies, study selection, charting the data, collating, summarizing, and reporting the results. The quality of included studies was appraised as recommended by Levac et al. [14]. The results of the scoping review are presented according to the Preferred Reporting Items for Systematic and Meta-analysis Extension for Scoping Reviews (PRISMA-ScR) [15].

## Eligibility of research question

The study research question was: What is the sustainability of REASSURED POC diagnostic tests in SSA? Sustainability is defined as the ability to provide services at a level that will maintain the prevention and treatment for health matters after the termination of major financial, managerial as well as technical assistance from external donors [16].

To determine the eligibility of our research question for a scoping review, we used the population, concept, and context (PCC) nomenclature (Table 1).

## Identification of relevant studies

We conducted a comprehensive search for relevant published literature in the following databases: Scopus, Dimensions, ProQuest Central, Google Scholar, and EBSCOhost (MEDLINE, CINAHL, as well as AFRICA-WIDE). We also searched for grey literature including reports from government and international organizations such as WHO, FIND, and the FDA. We manually searched the reference lists of included articles to identify articles not included in the electronic databases. Language restrictions were not applied to prevent the exclusion of relevant articles. The search strategy was co-developed by the principal investigator (PI), a subject specialist, and faculty librarian to ensure the correct use of keywords and Medical Subject Headings (MeSH) separated with Boolean operators [AND] and [OR]. The following keywords were used: point-of-care diagnostics, point-of-care testing, acceptability, SSA, and REASSURED. Keywords were refined to suit each database. Each search was documented in detail showing the keywords and MeSH terms, date of search, electronic database, as well as the number of publications retrieved (S1 Table).

## Study selection

Relevant studies were selected using the following inclusion and exclusion criteria:

**Inclusion criteria.** Articles were included in the scoping review if they met the following criteria:

- Articles reporting on all POC diagnostic tests used at all levels of healthcare systems

- Articles lacking comparative evidence between all levels of healthcare systems

**Table 1. Population, concept, context for determining the eligibility of the research question.**

| Population | Point-of-Care (POC) diagnostic tests |
|---|---|
| Concept | REASSURED (Real-time connectivity, Ease of sample collection, Affordable, Sensitive, Specific, User-friendly, Rapid, Equipment-free, Delivered) |
| Context | sub-Saharan Africa |

- Articles reporting on REASSURED diagnostics

- Articles published since the inception of REASSURED POC diagnostic tests

   **Exclusion criteria.**   Articles were excluded if they met the following criteria:

- Articles on laboratory-based POC diagnostics

- Review articles

All retrieved articles were screened in three stages. First stage: The titles of retrieved articles were all screened by BM. Eligible studies were exported to an Endnote 20 library and duplicates were removed. Second stage: Two reviewers (BM and BL) screened the article abstracts in parallel. The reviewers discussed discrepancies in selected articles until they agreed on the inclusion of articles. Third and last stage: All relevant full articles were screened by the two reviewers. Discrepancies in reviewers' responses following full article screening were resolved by a third reviewer (SRN). The level of agreement between the reviewers' feedback from abstract and full article screening was determined by calculating Cohen's kappa statistic. A kappa statistic value < 0.10 was considered as no agreement. Kappa statistic values between 0.10–0.20 indicated no to slight agreement, 0.21–0.40 indicated fair agreement, 0.41–0.60 indicated moderate agreement, 0.61–0.80 indicated substantial agreement, and 0.81–1.00 indicated almost perfect agreement.

## Charting the data

Data were captured from the included articles using a data charting form. Two independent reviewers (BM and TMT) piloted the data charting form and recommended modifications which were implemented. The following data were extracted from the included articles: author and publication year, title of study, aim of the study, study population, study setting, type of REASSURED POC criteria investigated, study findings, and conclusions.

## Collating, summarizing, and reporting results

We thematically analyzed data extracted from all included studies. The themes were narratively summarized.

## Quality appraisal

We used the mixed method appraisal tool (MMAT) version 2018 to assess the quality of the included articles [17]. BM and KM carried out the quality assessment process and calculated the overall percentage quality score. The scores were graded ranging from ≤ 50% to 100%. Percentage quality scores ≤ 50% represented low quality evidence; 51–75% represented average quality evidence, and 76–100% represented high-quality evidence.

## Ethical consideration

This scoping review did not include human or animal participants and was not completed for degree purposes, thus ethical approval was not required. All included and excluded studies following full article screening were cited.

## Results

### Screening

We screened the titles of 138 articles comprising 134 research articles and four articles from grey literature (S1 Table). We identified and removed 47 duplicates and subsequently screened the abstracts of 91 articles. During abstract screening, 24 articles met the inclusion criteria for full-article screening, and 67 articles were excluded. Full-article screening led to the inclusion of five articles and exclusion of 19 articles which did not meet the inclusion criteria (Fig 1). Articles were excluded due to the following reasons: reported on laboratory-based POC diagnostic services [18–27], were not conducted in SSA [18,24,26–29], commentary article [30], and focused on the wrong population [27,28].

Data was extracted from the remaining five articles. The level of agreement was calculated after full-article screening (S3 Table). A moderate agreement of 76% from the expected 56.16% was obtained (Kappa statistics = 0.4526, p-value < 0.05). Furthermore, the McNemar's chi-square statistics suggested that there was no statistically significant difference in the proportions of yes/no answers by the reviewers (p-value > 0.05).

### Characteristics of included studies

The characteristics of included studies are summarized in Table 2. The included studies were published between 2017 and 2022.

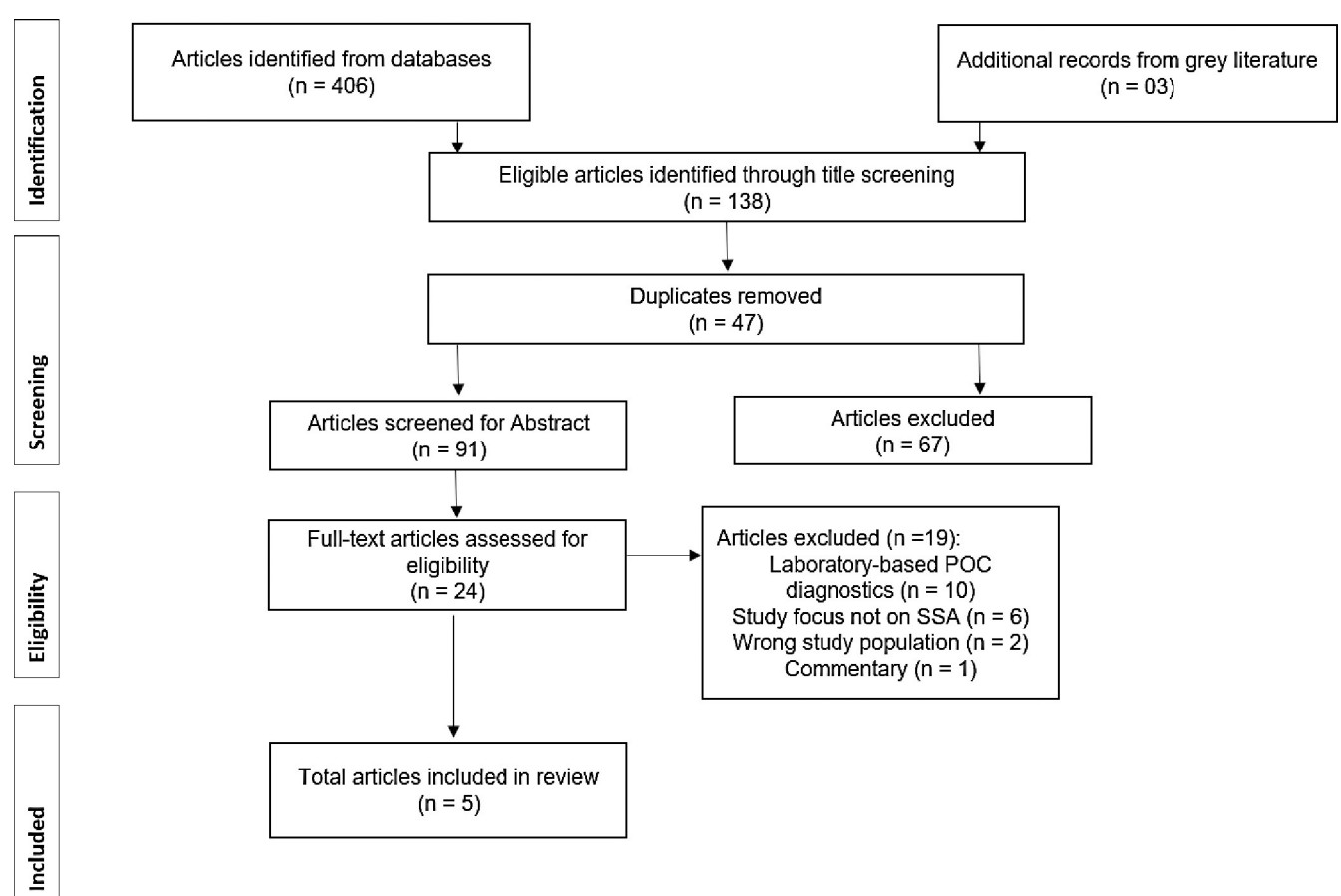

**Fig 1. PRISMA-ScR flowchart showing literature screening and selection of eligible articles.**

**Table 2. Characteristics of the included studies in the scoping review addressing the acceptability of REASSURED diagnostics in sub-Saharan Africa.**

| Author & Year | Title | Aim | Country | Design | Setting | Population | Type of POC test | REASSURED criteria assessed |
|---|---|---|---|---|---|---|---|---|
| Kimani, Mwangi [31] | Rethinking the design of low-cost Point-of-Care diagnostic devices | Describe the perceptions of Kenyan healthcare workers regarding the acceptability of rapid diagnostic tests (RDTs). | Kenya | Survey | Low-and middle-income (LMIC) | Kenyan healthcare workers | Malaria, HIV, syphilis, typhoid, pregnancy RDT. | Affordability |
| Land, Boeras [9] | REASSURED diagnostics to inform disease control strategies, strengthen health systems and improve patient outcomes. | Assess factors contributing to the success and failure of ASSURED diagnostics. | South Africa, Amazon forest, rural China | Evaluation study | LMIC | General population | HIV, malaria, syphilis, dual HIV and syphilis RDTs, TB lipoarabinomannan (LAM) antigen test. | Real time connectivity, ease of sample collection, affordability, sensitivity, specificity, user-friendliness, rapid and robust, equipment-free, delivered |
| Smith, Land [32] | Printed functionality for point-of-need diagnostics in resource-limited settings | To summarize important developments in micro-and nano-technologies for point-of-need health and environmental diagnostics. | South Africa | Evaluation study | LMIC | General population | Glucose monitoring home test, lateral flow test strips. | Real time connectivity, ease of sample collection, affordability, sensitivity, specificity, user-friendliness, rapid and robust, equipment-free |
| Turbé, Herbst [33] | Deep learning of HIV field-based rapid tests | Explore the potential of deep learning algorithms to classify RDT images as either positive or negative, focusing on HIV in KwaZulu-Natal. | South Africa | Cohort Study | LMIC | Healthcare workers | HIV RDT | Real time connectivity, sensitivity, specificity, user-friendliness |
| Dexter and McGann [34] | Saving lives through early diagnosis: The promise and role of point of care testing for sickle cell disease | Summarize the state of POC tests for sickle cell disease (SCD). | Ghana, Nigeria, India, Haiti, Tanzania, Mali, Togo | Cross-sectional study | LMIC | General population | Sickle cell disease (SCD) lateral flow immunoassay | Real time connectivity, ease of sample collection, affordability, sensitivity, specificity, user-friendliness, rapid and robust, equipment-free, delivered |

Geographic distribution of included studies showed South Africa to have most of the evidence on REASSURED POC diagnostic tests among SSA countries (Fig 2). Two POC test field evaluation studies [9,32] and one cohort study were conducted in South Africa [33]. One survey study was conducted in Kenya [31]. A multi country cross sectional study was conducted in Ghana, Mali, Tanzania, and Togo [34].

Our scoping review showed evidence on the following POC diagnostic tests: HIV rapid diagnostic tests (RDTs) [9,31,33]; dual HIV and syphilis RDTs [9]; glucose monitoring home test [32]; lateral flow test strips [32]; malaria RDT [9,31]; pregnancy RDT [31]; syphilis RDT [9,31]; sickle cell disease (SCD) lateral flow immunoassay POC test [34]; TB lipoarabinoman-nan (LAM) antigen test [9]; and typhoid RDT [31]. Fig 3 indicates the various POC tests in the included studies.

The identified studies provided evidence on at least one criterion of a REASSURED POC diagnostic test. The REASSURED criteria assessed were real-time connectivity [9,32–34], ease

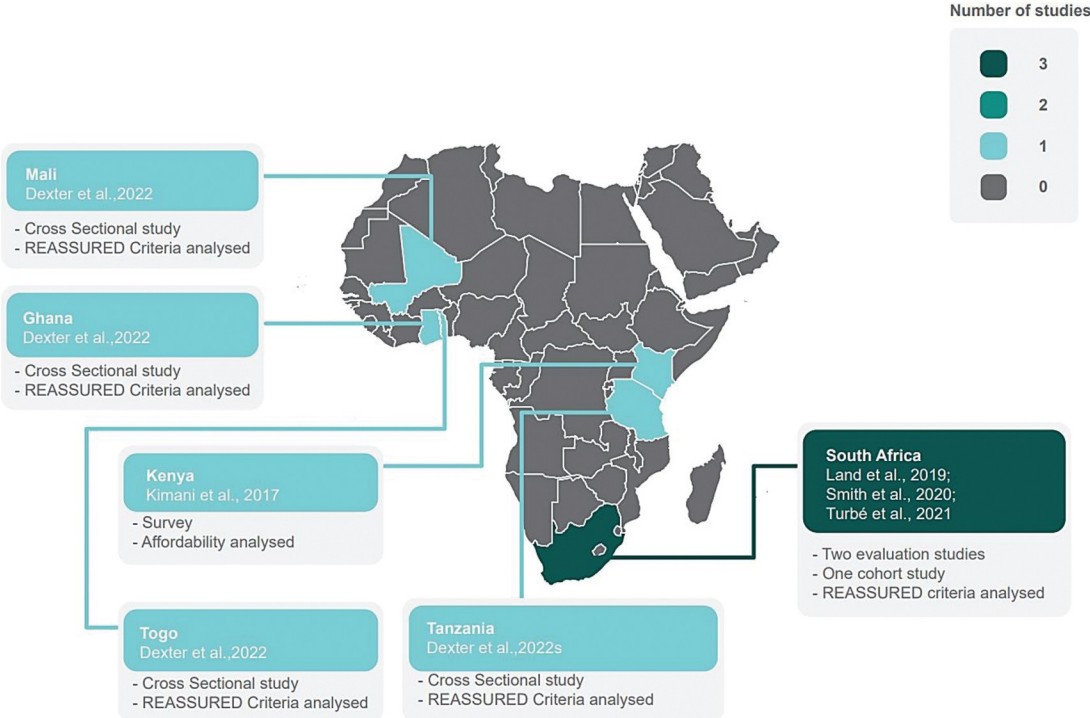

**Fig 2. Map of Africa showing a geographic distribution of research evidence on included REASSURED point-of-care diagnostic tests in sub-Saharan Africa.** Please click on link for base layer of the map: https://mapsvg.com/static/maps/geo-calibrated/africa.svg.

of sample collection [9,32,34], affordability [9,31,32,34], sensitivity [9,32–34], specificity [9,32–34], user-friendliness [9,32–34], rapid and robust [9,32,34], equipment-free [9,32,34], and delivered [9,34]. Fig 4 indicates the REASSURED criteria assessed from the eligible studies.

## Quality of evidence

The five eligible studies that were appraised for methodological quality scored between 90% and 100%, which indicated high methodological quality. Two studies had quality scores of 100% [9,31], two studies scored 95% [32,34] and the remaining study scored 90% [33] (S2 Table).

## Main findings

All the articles presented evidence on the acceptability of at least one REASSURED POC test criteria in SSA. The following themes emerged from the included articles: quality assurance on accuracy of REASSURED POC diagnostic tests, sustainability of REASSURED POC tests, and local infrastructure capability for delivering REASSURED POC tests to end users.

**Quality assurance on accuracy of POC diagnostic tests.**   Three studies reported on the quality assurance of POC diagnostic tests [9,31,33]. These studies highlighted the role of quality assurance on the reliability (false positives/or false negatives) of POC diagnostic tests in SSA [31]. Kimani, Mwangi [31] indicated that 50% of respondents felt that the reliability of POC diagnostic tests was a major obstacle for clinicians and that unreliable tests may lead to the incorrect prescription of medication. Land, Boeras [9] highlighted the importance of regular quality assurance updates to improve the implementation of POC tests. These quality assurance efforts include developing key policies and quality documents which healthcare workers

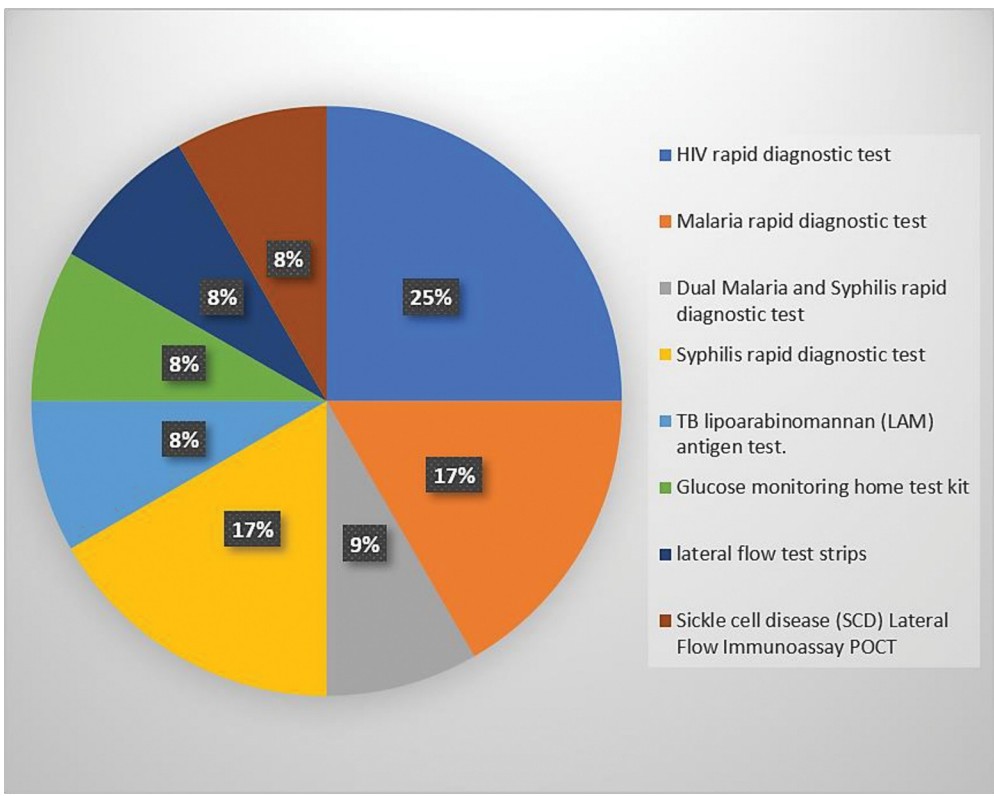

**Fig 3. List of point-of-care diagnostic tests sub-Saharan Africa.**

## REASSURED criteria assessed in eligible studies

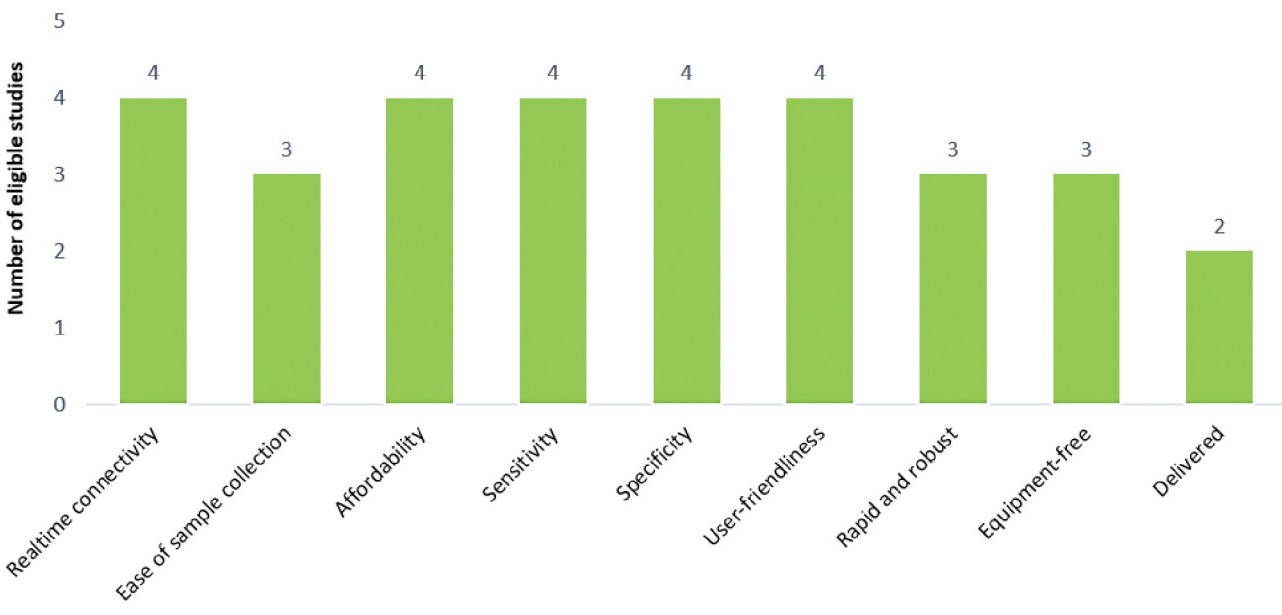

**Fig 4. REASSURED criteria assessed in the articles included in the REASSURED diagnostics at point-of-care in sub-Saharan Africa.**

need to be familiar with to ensure proper handling and interpretation of tests [9]. Turbé, Herbst [33] demonstrated the potential of deep learning for accurate classification of RDT images with an overall performance of 98.9% accuracy which was higher than the accuracy of traditional visual interpretation of 92.1%. The number of false positives recorded in the mobile-Health (mHealth) system were lower than the false positives reported for traditional visual interpretation. Though the accuracy increase was not significant even a slight increase in quality assurance could impact the lives of millions of people. Their study further laid a foundation for deep learning-enabled REASSURED diagnostics, demonstrating that RDTs linked to a mobile device could standardize the capture and interpretation of results for decision makers thus reducing errors [33]. Their findings presented evidence on the quality assurance of REASSURED POC diagnostic tests with the focus on accuracy but they did not assess the impact of healthcare workers' competency on the accuracy of REASSURED POC diagnostic tests in SSA.

**Sustainability of REASSURED POC tests.** Four studies reported on the sustainability of REASSURED POC diagnostic tests. Land, Boeras [9] evaluated how diagnostics can be improved by incorporating new technological elements which provide real-time quality control for testing and treatment. They highlighted how smart devices could be used to automatically add data to central databases allowing for diagnosis assessment and prediction of disease outbreaks [9]. Similarly, high levels of sustainability were shown for a smart device sending HIV RDT results to online databases in real-time [33]. Nurses and newly trained community health workers agreed that the devices were easy to use [33]. Kimani, Mwangi [31] surveyed clinicians and indicated that patients were satisfied with the results derived from RDTs (97%) and would recommend RDTs for use in the future (96%). They further recommended that devices should be convenient to user in order to ensure reliability, standardization, and ease of use [31]. The respondents also proposed the integration of technology developments and new knowledge into new diagnostic tests which are user-inspired [31]. Dexter and McGann [34] evaluated the performance of SCD POC tests and their importance in LMICs. Hemo Type SC test performed well in a Nigerian pilot study at immunization centers. Approximately 70% of screen-positive infants were enrolled in specialist care programs [34]. Field studies suggest that POC tests are important in improving healthcare delivery and outcomes. We could not find any evidence on the cost-effectiveness of implementing POC diagnostic tests in SSA healthcare systems.

## Local infrastructure capability for delivery to end user

One study reported on the lack of local infrastructure for production and distribution of REASSURED POC diagnostic tests. Smith, Land [32] presented research conducted on the development of printed wireless connectivity with low-cost sensing capabilities as part of a health diagnostic solution. The developed solution demonstrated vast low-cost substrates for paper-based diagnostic tests [32]. The study further reported the need to establish infrastructure and equipment for developing point-of-need solutions in South Africa for improved quality of life [32]. The findings presented evidence on the small-scale development of POC diagnostic tests which adhere to the REASSURED criteria, but we could not find any evidence on the competency of SSA manufacturers, researchers, and distributors to achieve this intervention.

## Discussion

This scoping review mapped the evidence on REASSURED POC diagnostics with the goal of optimizing their acceptability in SSA. Our findings indicated that there is limited published

research conducted on REASSURED POC diagnostics and their sustainability in SSA. New developments in digital diagnostic technology have set the platform for the next generation of POC tests, which could catalyze transformation in healthcare delivery [10]. SSA countries may benefit from early adoption of these POC tests and accelerate progress towards Universal Health Coverage as well as the United Nations Sustainable Development Goal 3 (SDG) 3 [10]. There is a need for relevant stakeholders to work together in achieving these goals. There is also a need for quality assurance interventions such as training to manage the accuracy and subsequent availability of REASSURED POC diagnostic tests. To ensure the quality of POC tests, both skilled and non-skilled test handlers should be trained to use POC tests competently [7]. Accurate diagnosis will ensures that patients are treated correctly and that disease surveillance is trustworthy [35].

POC diagnostic tests have been demonstrated to be a time-effective intervention during healthcare emergencies. However, when cost analyses are done, these tests are observed to be cost-prohibitive [36]. In this scoping review, we revealed evidence evaluating the sustainability of REASSURED POC diagnostic tests in SSA [9,32,33], with most evidence coming from South Africa [9,32,33]. Data generated suggested a gap in interventions to prioritize REASSURED POC tests in primary healthcare for accommodation of under privileged patients who cannot afford lab-based diagnostic testing. Few studies have assessed the costs of using REASSURED POC diagnostics in SSA healthcare systems, particularly in remote areas. Primary studies assessing cost effectiveness could potentially assist in evaluating generated data and developing future studies that assess the sustainability of REASSURED POC devices in SSA. Our study further revealed the vast contrast in the amount of knowledge generated through primary studies from lab-based REASSURED diagnostics compared to studies generated for POC tests which are user-friendly for patients [18,20–24,29,37]. We could not retrieve any evidence on the SARS-CoV-2 virus REASSURED POC tests despite the scale of the pandemic.

A study in South Africa demonstrated developments towards REASSURED POC diagnostics using paper-based and printed electronic technologies [25]. We did not include this study in our scoping review because the evidence generated was for lab-based POC tests [25]. A similar study summarizing paper-based strip tests for blood glucose evaluated the ease of sample collection, specificity, as well as the sensitivity of the REASSURED criteria [26]. Optimization of the POC test strips for the semi-quantitative detection of glucose in whole saliva was demonstrated. However, this study was conducted outside SSA [26]. A previous study conducted in similar settings reported on the development of high performance POC tests. The study evaluated a low-cost, highly sensitive, automated, and robust paper/soluble polymer hybrid-based lateral flow platform paired with a smartphone reader for application in resource-limited settings [20]. With the movement of POC test development and manufacturing to smart technology guided by the REASSURED criteria, their sustainability in SSA remains to be proven. Therefore, more SSA countries need to generate data on REASSURED POC diagnostic tests and address their sustainability with evidence-based interventions.

## Strengths and limitations

To the best of our knowledge, this is the first scoping review to provide a comprehensive overview of evidence on REASSURED POC diagnostic tests in SSA. Our scoping review employed a comprehensive database search which included empirical evidence with no language limitations. The retrieved articles were rigorously screened by two independent screeners. The screening tools were piloted and following full article screening, the kappa statistics calculation yielded a moderate agreement of 76% from the expected 56.16%. To ensure transparency, the included studies were subjected to quality appraisal using MMAT 2018 version. High

methodological quality of between 90–100% was obtained for the eligible studies. Despite the above strengths, this study has some limitations. This study was not limited to studies that were conducted in SSA. Forty percent of the included studies presented evidence from SSA and other countries outside Africa [9,34]. However, majority of countries presented in these included studies were from SSA.

## Suggestions for future practice

All the studies in this scoping review were conducted in SSA where access to high quality health systems is limited. From these studies, three [9,31,32] evaluated REASSURED POC diagnostic tests in SSA whilst two [33,34] were field work studies. This demonstrates the lack of evidence generated from local primary studies on POC diagnostic tests. Secondly, quality assurance, training as well as infrastructure were observed as barriers towards the sustainability of REASSURED POC diagnostics in SSA. Our findings indicate a need for local prioritization of research conducted to reflect the true acceptability of REASSURED POC diagnostic tests in SSA.

## Suggestions for future research

Our study revealed limitations in research conducted on REASSURED POC diagnostic tests in SSA. Our scoping review revealed a lack of primary studies on the training regime of field workers on quality assurance of REASSURED POC diagnostic tests during upscaling and implementation processes in SSA. Furthermore, the study revealed a lack of primary studies on the sustainability of REASSURED POC diagnostics by end users in remote regions of SSA. In addition to these gaps, the study revealed a lack of interventions have been developed in SSA to assess research competency, infrastructure readiness for manufacturing, and delivery of REASSURED POC diagnostic tests to end users. The gaps identified from the scoping review proposes future research in discovering training regimes for non-professional users of REASSURED POC diagnostic tests in order to maintain the quality in terms of accuracy of these tests. This will ensure proper patient treatment and enhance primary healthcare quality in resource-limited settings such as SSA. Considering challenges encountered with electricity supply and subsequent network failures, we further recommend future research to investigate the cost effectivity of REASSURED POC diagnostics on healthcare systems in remote regions of SSA. Future research can also evaluate the competency and readiness of SSA infrastructure for manufacturing, packaging and delivery of REASSURED POC diagnostic devices to end users.

## Conclusions

There is limited research on REASSURED POC diagnostics and their sustainability in SSA. Quality assurance of POC tests is essential in ensuring patients get accurate results and are prescribed proper treatment. The global movement of POC tests to digital technology makes these REASSURED devices important at all levels of healthcare for proper surveillance and treatment of communicable and non-communicable diseases. The acceptability of REAS-SURED POC diagnostics in resource-limited regions such as SSA is thus required to enhance the quality of primary healthcare. The lack of local research studies towards the acceptability of REASSURED POC diagnostic tests in SSA delays the move towards improved quality primary healthcare delivery.

## Supporting information

**S1 Checklist. Preferred Reporting Items for Systematic reviews and Meta-Analyses extension for Scoping Reviews (PRISMA-ScR) checklist.**
(DOCX)

**S1 Table. Full search strategy in EBSCOHost.**
(DOCX)

**S2 Table. Methodological quality appraisal in included studies.**
(XLSX)

**S3 Table. Reviewers level of agreement: Full text screening (Yes = 1, No = 0).**
(DOCX)

## Acknowledgments

The authors would like to acknowledge the University of Pretoria Evidence Synthesis and Translation Research Group for their assistance with the scoping review. The authors would also like to extend their appreciation to the University of Pretoria Faculty of Health Sciences Library services for their assistance with optimizing the search strategy. Dr. Cheryl Tosh for editing.

## Author Contributions

**Conceptualization:** Boitumelo Moetlhoa, Tivani Mashamba-Thompson.

**Data curation:** Boitumelo Moetlhoa.

**Formal analysis:** Boitumelo Moetlhoa.

**Investigation:** Boitumelo Moetlhoa.

**Methodology:** Boitumelo Moetlhoa, Kuhlula Maluleke, Kabelo Kgarosi, Siphesihle R. Nxele, Bonolo Lenonyane, Tivani Mashamba-Thompson.

**Supervision:** Tivani Mashamba-Thompson.

**Validation:** Tivani Mashamba-Thompson.

**Writing – original draft:** Boitumelo Moetlhoa.

**Writing – review & editing:** Evans M. Mathebula, Tivani Mashamba-Thompson.

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
