## [Decision Letter · Decision Letter 0]

27 Feb 2023

PGPH-D-22-01956

REASSURED diagnostics at point-of-care in sub-Saharan Africa: A scoping review

Dear Dr. Moetlhoa,

Thank you for submitting your manuscript to PLOS Global Public Health. After careful consideration, we feel that it has merit but does not fully meet PLOS Global Public Health’s publication criteria as it currently stands. Therefore, we invite you to submit a revised version of the manuscript that addresses the points raised during the review process.

We look forward to receiving your revised manuscript.

Kind regards,

Joao Tiago da Silva Botelho

Academic Editor

Journal Requirements:

2. Please provide separate figure files in .tif or .eps format only and remove any figures embedded in your manuscript file. Please also ensure that all files are under our size limit of 10MB.

3. Fig 2: please (a) provide a direct link to the base layer of the map (i.e., the country or region border shape) and ensure this is also included in the figure legend; and (b) provide a link to the terms of use / license information for the base layer image or shapefile. We cannot publish proprietary or copyrighted maps (e.g. Google Maps, Mapquest) and the terms of use for your map base layer must be compatible with our CC-BY 4.0 license. 

Additional Editor Comments (if provided):

Reviewers' comments:

Reviewer's Responses to Questions

**Comments to the Author**

1. Does this manuscript meet PLOS Global Public Health’s publication criteria? Is the manuscript technically sound, and do the data support the conclusions? The manuscript must describe methodologically and ethically rigorous research with conclusions that are appropriately drawn based on the data presented.

Reviewer #1: Partly

Reviewer #2: Yes

2. Has the statistical analysis been performed appropriately and rigorously?

Reviewer #1: Yes

Reviewer #2: Yes

3. Have the authors made all data underlying the findings in their manuscript fully available (please refer to the Data Availability Statement at the start of the manuscript PDF file)?

Reviewer #1: Yes

Reviewer #2: Yes

4. Is the manuscript presented in an intelligible fashion and written in standard English?

Reviewer #1: Yes

Reviewer #2: Yes

5. Review Comments to the Author

Reviewer #1: Dear Author,

This is a good work for giving evidence of REASSURED POC diagnostic tests in SSA. However, it is not enough information to persuade or give overview about the picture of diagnostic use with only 5 publications, even there is limitation of publication.

If the author just focus on performing the test at point of care, the author should explain or give more factors contributing to explain the reason why the REASSURED diagnostic tests was rarely used in SSA, not just the quality of test, infrastructure...it may be the reason from cost, the supplier or supply chain management framework, the variety of diagnostic tests - no eligible tests for some diseases, policy, accessibility of POC testing services...if the author mention about the quality of test including sensitivity and specificity or quality assurance that was a part of lab activities. Unfortunately, the author excluded all publication related to lab works.

When discussing about the sustainability of REASSURED POC diagnostic, not just perform at near the side of patient from non - laboratory professional, but there are many factors effect to it such as the access to quality diagnostic tests as well.

Reviewer #2: I have read this scoping review on mapping ‘REASSURED diagnostics at point-of-care in sub-Saharan 3 Africa’ with great pleasure. The paper is well written, and I feel it has great potential in contributing to further research in the field of infectious diseases.

Despite these merits, here are some observations and comments for your consideration

1. I thought the study was focused on SSA. Kindly clarify lines 311-312 (strengths and limitation section) that state ‘This study was not limited 312 to studies that were conducted in SSA’. See the scoping review question.

2. Kindly define sustainability. Clarify the focus of the scoping review.

3. Great you mentioned SARs CoV 2 virus was not identified, would international adding SARs CoV 2 to the search string make a difference especially if you extended the search to late 2022? OR could there be reasons why REASSURED was not applied to SARs CoV 2 virus?.

4. Following the research question, the inclusion criteria would include:`

a. Articles reporting REASSURED in SSA papers, Or in exclusion, you state non-SSA studies were not included, and therefore would have benefited from Boolean NOT or what would this mean for the study in SSA, china etc., and how did not included this

Main findings

• Need clarification on sustainability and reframing.

• Discussion section

• Need clarification: Lines- 273/ 286-287: Start a discussion first, then recommendations.

• On the same note, lines: 288-290 while this information is important, you clearly indicated you will not focus on this, I, therefore do not understand why this is being discussed here.

• Literature introduced in the discussion section should be included in the introduction section.

6. PLOS authors have the option to publish the peer review history of their article (what does this mean?). If published, this will include your full peer review and any attached files.

**Do you want your identity to be public for this peer review?** For information about this choice, including consent withdrawal, please see our Privacy Policy.

Reviewer #1: **Yes: **Diep The Tai

Reviewer #2: **Yes: **Emmy Kageha Igonya

---

## [Decision Letter · Decision Letter 1]

12 May 2023

REASSURED diagnostics at point-of-care in sub-Saharan Africa: A scoping review

PGPH-D-22-01956R1

Dear Dr Moetlhoa,

We are pleased to inform you that your manuscript 'REASSURED diagnostics at point-of-care in sub-Saharan Africa: A scoping review' has been provisionally accepted for publication in PLOS Global Public Health.

Best regards,

Joao Tiago da Silva Botelho

Academic Editor

Reviewer Comments (if any, and for reference):

Reviewer's Responses to Questions

**Comments to the Author**

1. If the authors have adequately addressed your comments raised in a previous round of review and you feel that this manuscript is now acceptable for publication, you may indicate that here to bypass the “Comments to the Author” section, enter your conflict of interest statement in the “Confidential to Editor” section, and submit your "Accept" recommendation.

Reviewer #1: All comments have been addressed

Reviewer #2: All comments have been addressed

2. Does this manuscript meet PLOS Global Public Health’s publication criteria? Is the manuscript technically sound, and do the data support the conclusions? The manuscript must describe methodologically and ethically rigorous research with conclusions that are appropriately drawn based on the data presented.

Reviewer #1: Partly

Reviewer #2: Yes

3. Has the statistical analysis been performed appropriately and rigorously?

Reviewer #1: Yes

Reviewer #2: N/A

4. Have the authors made all data underlying the findings in their manuscript fully available (please refer to the Data Availability Statement at the start of the manuscript PDF file)?

Reviewer #1: Yes

Reviewer #2: Yes

5. Is the manuscript presented in an intelligible fashion and written in standard English?

Reviewer #1: No

Reviewer #2: Yes

6. Review Comments to the Author

Reviewer #1: Accept

Reviewer #2: The topic under review is important. I'm please the authors have addressed all comments appropriately, and through sustainability, they have provided more information beyond just mapping the implementation of REASSURED.

7. PLOS authors have the option to publish the peer review history of their article (what does this mean?). If published, this will include your full peer review and any attached files.

**Do you want your identity to be public for this peer review?** For information about this choice, including consent withdrawal, please see our Privacy Policy.

Reviewer #1: **Yes: **Diep The Tai

Reviewer #2: **Yes: **Emmy kageha igonya
